# Against All Odds: Tentative Steps toward Efficient Information Sharing in Groups

**Darius Schlangenotto [1], Wendelin Schnedler [2,3,4],\*  and Radovan Vadovič [5]**

[1] Business Information Systems Department, Faculty of Business Administration and Economics, Paderborn University, D-33098 Paderborn, Germany; darius.schlangenotto@upb.de

[2] Management Department, Faculty of Business Administration and Economics, Paderborn University, Warburger Straße 100, D-33098 Paderborn, Germany

[3] Centre for Market and Public Organization, University of Bristol, Bristol BS8 1QU, UK

[4] Institute of Labor Economics (IZA), Schaumburg-Lippe-Straße 5–9, 53113 Bonn, Germany

[5] Economics Department, Carleton University, Ottawa, ON K1S 5B6, Canada; radovan.vadovic@carleton.ca

\* Correspondence: wendelin.schnedler@upb.de

**Abstract:** When groups face difficult problems, the voices of experts may be lost in the noise of others' contributions. We present results from a "naturally noisy" setting, a large first-year undergraduate class, in which the expert's voice was "lost" to such a degree that bringing forward even more inferior information was optimal. A single individual had little chance to improve the outcome and coordinating with the whole group was impossible. In this setting, we examined the change in behavior before and after people could talk to their neighbors. We found that the number of people who reduced noise by holding back their information strongly and significantly increased.

**Keywords:** information aggregation; coordination; communication; swing voter's curse

## 1. Introduction

People have varying degrees of expertise and views of the facts. Nevertheless, they often need to come to a decision without the help of some leader who imposes her view. Is the developed product ready to be launched? Should the UK leave the EU? While people are free to discuss facts and what to make of them with a colleague or a friend, the joint time of all involved is a scarce resource, especially if the group is large. Not every team member can elaborate on the chances of a successful launch. Asking every UK citizen to describe the perceived benefits and costs of leaving the EU and then agreeing on what these contributions imply seems absurd. Instead, communication is restricted—in the extreme case to an expression of opinion by a vote on a proposal that reduces potentially rich information to the rather coarse "yea," "nay," or "nil."

When group communication is restricted, holding back less valuable information can become optimal—a notion that has been formalized by Feddersen et al. [1] as the "swing voter's curse." Talking about a minor detail in a team meeting may bury potentially important contributions of others. Voting on whether to leave the EU based on one's gut feeling dilutes the voices of those who correctly foresee the implications and consequences of such a step. Individuals do not always realize that withholding information may improve the group's decision. In practice, people frequently talk or vote without much to go on. For such groups to improve information aggregation, individuals would have to learn about the benefits of restraining themselves and let the expert be heard.

While direct communication can help small groups in coordinating for better outcomes, (for examples, see [2–5]) such coordination may be hampered by three features in our initial examples.

First, a large group size directly complicates coordination. Second, if others are unaware of the efficient strategy, trying to coordinate on this strategy with them seems less promising. Third, coordination is difficult if those involved cannot talk to each other. Communication may even be detrimental when it is local. Someone who has the insight to restrain herself may realize, while conversing with others, that ignorance is prevalent. If she believes that group behavior is unlikely to change for the better, it becomes optimal for her to no longer restrain herself and contribute the little that she knows. On the other hand, local communication may foster learning and people may trust that others are independently reaching similar conclusions and restrain themselves. Can a large group learn to withhold inferior information? Is local communication helpful in achieving this goal?

Answering these questions in the field is difficult because we cannot observe the quality of information that is withheld. In an experimental study, however, the quality of information can be controlled by the experimenter. We opted to run the experiment in a first-year class with more than 500 participants. We did so to capture the three above-mentioned features of real-life situations. First, we can have several groups of 36 members jointly deciding on an issue, while not being able to reach out to one another and coordinate because they are randomly dispersed across the auditorium. In an economics laboratory that seated 20–30 people, this would have been very hard to do. Second, while subjects in the laboratory seem to be aware of the benefits of holding back to such a degree that Morton et al. [6] speak of a norm of "letting the experts decide," our experience suggested that participants in a large lecture might not. Third, the classroom setting allows us to have "rich" local communication, while that on the group level is restricted to voting: students sit next to those whom they are likely to know and trust, but they decide together with other students who are scattered across the room.

One out of the 36 group members received expert knowledge: she was perfectly informed about the most frequent color in an urn with blue and green balls. For the other 35, we drew balls from the urn without replacement and showed them the colors of their individual draws on a mobile device. In order to introduce some uncertainty about the value of one's own information, one of the 36 individuals was randomly excluded from the group. The perfectly informed individual was thus not certain but very likely to remain in the group. Every remaining individual could then either abstain or vote for or against the color that they saw. If the majority voted for the right color, all in the group received €10. Otherwise, they got nothing.

This game featured an *all-vote equilibrium*, where everybody votes for their information, but also an *only-expert-votes equilibrium*, in which everyone but the expert abstained. We parameterized it so that the only-expert-votes equilibrium was substantially more efficient: the probability of identifying the correct state was more than 30 percentage points higher than when all voted. Our game is similar to that by Morton et al. [6], so our experiment can be regarded as an extension and robustness check bringing in the above features.

As we expected and hoped for, participants initially overwhelmingly voted for their own color (78.1%). This is not only a coordination problem. In a post-experimental questionnaire, we asked participants the hypothetical question: If you would interact with a group of robots, how would you program them? More than half of the people programmed the robots to always vote and only a quarter to play the efficient strategy. This suggests that they were unaware of the efficient strategy. Moreover, participants were more likely to vote against their own color (15.5%) than to abstain (6.4%). Given this behavior, voting one's own color is optimal.

Taking inspiration from Cason et al. [7], who found that local communication among students reduces misconceptions in an individual decision task, we gave students five minutes to talk to their immediate neighbors before repeating our interactive decision task with a newly filled urn and in new teams. While individual changes in voting behavior had little impact, a coordinated response by the group could improve its performance by five percentage points if the few people who voted against their color and abstained could be convinced to vote in line with their color. Alternatively, the group could obtain a gain of over 40 percentage points if almost everybody in the group but the

expert abstained. As pointed out, voting one's own color is the best response to the initial behavior of others. Given that direct communication with the other group members was not possible, these circumstances were unlikely to change and an increase in voting thus seemed likely.

Still, we found that abstentions increased in the second run from 6% to 13%. This increased by more than 100% was not only economically but also highly statistically significant. It documents, to our knowledge for the first time, that self-governed groups are not trapped but make a tentative step toward the more efficient use of information. Put differently, we may have been observing the birth of a "norm of abstention". The increase is all the more surprising because it is (predictably) too small to improve information aggregation: if anything, the majority is now less likely to identify the right color.

One possible explanation for why abstentions increase is that participants teach each other about the sophisticated strategy. Consistent with this notion, people who program the sophisticated strategy are significantly more likely to start abstaining in the second round (86%) than those who do not (33%). Moreover, abstention spreads locally. With a neighbor who abstained in the first round the probability that a voter abstains in the second round increases by 13%.

In our setting, communication occurs naturally, without interference, structure, or observation. The downside of this is that we are unable to use the content of conversations to decide whether people really learn about the sophisticated strategy or whether they already know this strategy and only learn to trust that coordination can be successful. It is even possible that the increase in abstention had nothing to do with communication at all but instead resulted from locally correlated independent learning. More able participants are more likely to find the sophisticated strategy either in the first or the second round than less able ones. If people of similar ability are more likely to sit next to each other, having a neighbor who abstained in the last round is associated with a higher probability to switch to abstention in the second round—even without communication. We cannot exclude this explanation because participants were not seated randomly but (as is typical for lectures) chose their seats themselves.[1] Our design can also not exclude that people learn to trust in successful coordination. We can, however, examine whether these alternative explanations are consistent with the data.

Both alternative explanations, local but independent learning and learning to trust in the success of coordination, imply that having more abstaining neighbors in the first round is associated with a stronger increase in abstentions in the second round. The more neighbors of an individual have abstained in the previous round, the more likely appears a successful coordination to that individual. She becomes more willing to abstain. A participant with more able independent learners around is more likely to discover the sophisticated strategy and to abstain.

These alternative explanations, however, are *not* borne out by the data. How many neighbors previously abstained is not associated with a higher probability to abstain. The explanations of local independent learning and learning to trust in the success of coordination are thus not consistent with what we observed. Our tentative conclusion is that people teach each other the sophisticated behavior—although this is lowering the group's probability of success.

The remainder of the paper is organized as follows. The next section describes our contribution to the literature. Section 3 derives the best-reply and some equilibria of the game. Section 4 explains the design of the study. Section 5 presents our findings and Section 6 concludes.

## 2. Contribution to the Literature

Our paper relates to three strands of the literature: (i) optimal extraction of information in groups, (ii) choice of equilibrium when there is communication, and (iii) learning in experiments.

The proposition that more people know more has fascinated social philosophers at least since Galton conducted his famous analysis showing that the average guess of visitors at a fair was closer

---

[1]　　Seating this amount of people randomly and checking that they sit in the designated seats would not have been possible in the allocated time.

to the true weight of an ox than that of an expert [8]. De Condorcet [9] asserts that more (at least partially) independent and informative viewpoints lead to better decisions. Here, we check whether crowds are not only wise in the sense of available information but also whether they wisely aggregate this information.

How groups aggregate information has received quite a bit of attention in the literature. One of the best established results in this regard is that markets work extremely well in extracting and pooling private information from individual traders. This power of markets to collect and disseminate information has been demonstrated by Plott et al. [10]. In the presence of complete markets,[2] the price swiftly converges to the rational expectation's benchmark. Plott et al. [11] and Camerer et al. [12] examined the role of "expertise" (or perfectly informed insider) in experimental asset markets. They show that the experts' information quickly takes a hold and the prices converge to the underlying fundamental value.[3] In contrast, information is shared in our study by voting.

There is a sizable literature examining information aggregation when voting. For instance, Guarnaschelli et al. [14] examined the strategic voting incentives in ad hoc committees of various sizes under the majority and unanimity rule. Ali et al. [15] further extended this work to standing committees that interact repeatedly. In line with assertions of Austen et al. [16] and Feddersen et al. [1], these studies found that strategic voting is prevalent. Under the unanimity rule, as predicted, a substantial fraction of subjects vote against their signal.[4] These findings demonstrate that in laboratory voting games, subjects are capable of acting with a high degree of sophistication.[5] In our case, there are no incentives to vote against one's own signal. On the other hand, strategic abstention is central to the efficient information aggregation strategy that we focus on.

Unlike in our study, a high degree of coordination on abstention by non-experts is observed in laboratory experiments. Exploring Feddersen et al. [1,20] in a series of experiments, Battaglini et al. [21,22] and Morton et al. [6] found that a large fraction of subjects withheld their information if there was a high chance of a better informed expert. In Morton et al. [6] some learning of abstention seemed to be going on, which suggested that we might see the emergence of a "norm of abstention." This, however, was by no means certain. In all these experiments, standard procedures ensure common knowledge of the game and the problem that we are interested in studying, i.e., whether abstention can propagate in a group where many are unaware of the value of abstention, does not arise.

The sophistication of individuals when extracting information in common value environments can reach its limits. If public and private information is available, members of a group can, for example, be prone to updating biases. In Mengel et al. [23], inefficiencies arise because subjects trust available expert information too much and then vote too often against their signal. In Kawamura et al. [24], subjects are locked in a situation where they vote too often for their own uninformative signal rather than abstain. Following Charness et al. [25], who inquired into whether beliefs or cognitive difficulty are at the heart of overbidding in auctions, Esponda et al. [26] addressed this issue in a voting context and identified a subject's difficulty to engage in hypothetical thinking as the root of non-strategic voting.[6] Neither of these contributions examines how behavior develops if subjects have the opportunity to communicate and hence to learn from each other.

The second strand of literature to which we relate is coordination between differently efficient equilibria when there is communication. A sizable literature documents the positive impact of pre-play communication on the ability of the group to coordinate on Pareto efficient equilibrium; e.g., Cooper et al. [2], Charness [3], Duffy et al. [4], Blume et al. [5], Cason et al. [28], and Blume et al. [29].

---

[2]   For instance, prediction markets that use Arrow–Debreu securities.
[3]   For an excellent survey of the literature, please see Plott [13].
[4]   Strategic voting is a common behavioral trait in laboratory experiments. For a nice survey, see Martinelli et al. [17].
[5]   Somewhat weaker evidence for a fully rational benchmark is found in experiments with costly information acquisition, e.g., Elbittar et al. [18] or Bhattacharya et al. [19], where subjects are typically found to over-invest in information.
[6]   The same reasoning was examined in the context of markets by Ngangoue et al. [27].

Goeree et al. [30] examined the role of free communication on the behavior in a subsequent experimental voting game. They found that subjects not only truthfully and publicly revealed their private information to each other but they also discussed what to do with that information (i.e., how to vote) to achieve the best outcome.[7] This decentralized process uniformly improves group decision making and diminishes the role of strategic incentives presented by different voting institutions. Palfrey et al. [34] found a significant positive impact of intra-party communication on turnout in laboratory elections. Contrary to Nash equilibrium predictions, the higher turnout primarily benefits the majority party.[8] In both these contributions, communication was public (within the relevant group of voters) and could thus be used to establish common knowledge. In our study, we were interested in a situation wherein communication does not extend to the whole group, so that common knowledge cannot be achieved.

Finally, our paper offers a contribution to learning in experiments. Various experiments examine how subjects learn individually in interactive situations, e.g., in Cournot or Bertrand competition (see, e.g., [36,37]), social-dilemma games (see, e.g., [38–40]), and coordination games [41,42]. In other contributions, subjects learn from others in individual decision problems [7,43,44]. We looked at a situation in which individuals could learn from others but in an interactive situation.

## 3. Model and Analysis

The game that is played by participants involves 36 players $i \in \{1, \ldots, 36\}$. Each player receives a signal $s_i \in \{\text{blue, green}\}$ about an unknown state of nature $\theta \in \{\text{blue, green}\}$, which represents whether the majority of balls in an urn are blue or green. Signals are of quality $t_i$. One of the signals is perfect $t_i = 1$ and reveals the true state $P(s_i = \theta | t_i = 1) = 1$. All others are relatively noisy because they are drawn from the urn which contains 99 balls of which 50 have one color and 49 the other color. One randomly drawn player is excluded, so that 35 players remain and it is very likely that one of these is perfectly informed ($\frac{35}{36} \approx 97.2\%$).

The remaining 35 players can vote blue, green or abstain depending on the content $s$ and type $t$ of the signal. Once signals are realized, strategies result in a voting outcome $v$, where $v$ measures how many more people voted for the true than the other state $v \in \{-35, \ldots, 0, \ldots 35\}$. Let us standardize the payoff in the case that the majority voted for the actual state, $v > 0$, to one and for $v \leq 0$ to zero.

Player $i$'s vote is decisive in two situations: in the case of a tie, $v_{-i} = 0$, and in the case of a one-vote lead for the true state, $v_{-i} = 1$, where $v_{-i}$ is the voting outcome without $i$'s vote. In the first case, a vote for the true state generates a majority for the true state and gains of one. In the second case, a vote against the true state destroys the majority and leads to losses of one. These situations may never occur. If, for example, all other players always vote blue irrespective of their signal, player $i$'s vote will never make a difference. In our setting of a large class, however, such a coordinated response is very unlikely. This is why we assume the following.

**Assumption 1** (Strategic uncertainty assumption). *From player i's perspective, it cannot be excluded that her vote affects results: $P(v_{-i} = 0) > 0$ and $P(v_{-i} = 1) > 0$.*

This strategic uncertainty means that player $i$'s behavior matters and gives her a reason to contemplate her choice in the first place. The assumption has several consequences. First, voting for one's own signal strictly dominates voting against it (see Lemma A1). The reason is that the signal is informative and voting against it is hence more likely to cause harm whenever the player can make a difference.

---

[7]　When interests conflict, communication has been shown to affect the willingness to give in a dictator game [31], to increase the power of a proposer in multilateral bargaining [32], but also to positively affect investments before sharing the gains from team production [33].

[8]　The role of local communication in an electoral process has recently been highlighted by Pons [35]. The field experiment involving French presidential elections showed that a short conversation may affect voting behavior.

Moreover, for perfectly informed players ($t_i = 1$) voting strictly dominates abstaining (see Lemma A2). Since they are perfectly informed, they will never destroy a one-vote lead for the true state by voting for their signal but they may resolve a tie in the right way.

For imperfectly informed players ($t_i = 0$), the situation is more interesting. On the one hand, voting for one's own signal can resolve a tie. On the other hand, it may destroy a one-vote lead. The gains from voting one's signal thus depend on the likelihood that these pivotal situations occur. The top panel in Figure 1 shows the gains from voting relative to abstaining under the assumptions that players with perfect information vote for their signal and given a probability $q$ that other imperfectly informed players (non-experts) vote. The figure actually contains both extreme cases, i.e., when non-experts vote for and against their signals. The difference, however, is so small that they do not show in the graph. The reason is that the expert is very likely to be present. Hence, the question whether others abstain or vote is more important than the relatively small difference in getting it right when voting.

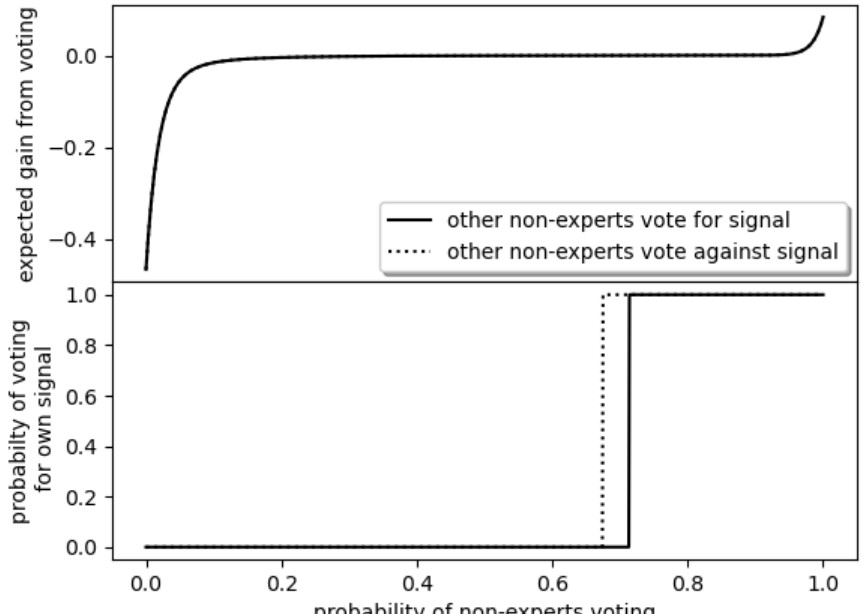

**Figure 1.** Expected gains from voting rather than abstaining and best response given the behavior of other non-experts.

Player $i$'s best reply in the lower panel shows that the likelihood that other imperfectly informed players vote is crucial. If it is unlikely, abstention is better to not dilute the expert's vote. If it is likely, the expert's vote is already so diluted that voting is the best reply.

The share for which it becomes optimal to vote rather abstain depends on the behavior of the other non-experts: if they all vote against their signal, voting for the signal becomes more valuable in comparison to abstaining and the share drops. The best reply function reveals the following equilibria of the game (the proof is in Appendix A).

**Proposition 1.** *Under the strategic uncertainty assumption, there are two Nash equilibria in pure strategies:*

- *In the only-expert-votes equilibrium, only the perfectly informed expert votes and all others abstain.*
- *In the all-vote equilibrium, everyone votes.*

*In addition, there is a Nash equilibrium in mixed strategies, wherein the expert votes and non-experts are likely to vote for their color (in around 70% of the cases) but sometimes abstain (in around 30% of the cases).*

Figure 1 shows that apart from extreme situations in which nearly everyone either abstains or votes, the individual decision to vote or abstain has very little impact on the group's probability of identifying the right color. The non-expert's joint decisions, however, can have a sizable impact (see Figure 2), so that it matters which equilibrium will be played (the proof is in Appendix A).

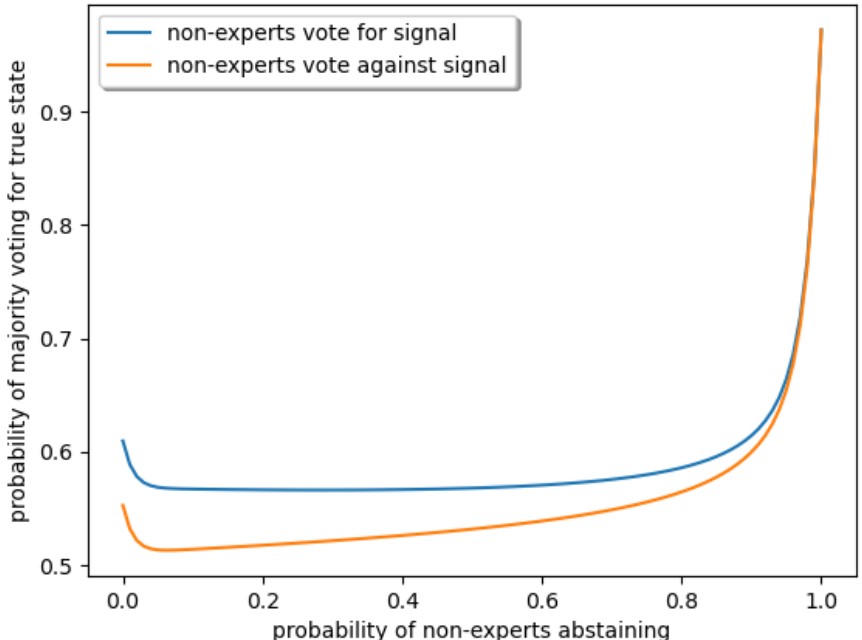

**Figure 2.** The group has the best performance if all but the expert abstains.

**Proposition 2.** *The only-expert-votes equilibrium Pareto-dominates the all-vote and mixed-strategy equilibrium by more than 30 percentage points.*

Since perfectly informed players vote in both equilibria, which of them is reached depends on the behavior of imperfectly informed players. This is why we focus on their behavior when discussing results, later.

## 4. Experimental Design

When introducing the game in the lecture, we used physical props to render it easy for subjects to comprehend the problem. Before the lecture, we filled two urns with 49 blue and 49 green balls, and added another (blue or green) ball to generate a majority and a dice of the same color (to represent the perfect information). We then drew the dice and 35 balls from the urn, numbered them, and placed them into a small cardboard box that was sealed and put on display during the whole lecture. After the complete experiment, the box was opened and students were allowed to inspect them.

Each participant in the lecture theater received the information about the color of one object from the cardboard box on an Internet capable device,[9] i.e., a mobile phone, tablet, or laptop. The ball (imperfect information) was represented on the screen by a circle of the appropriate color; the dice (perfect information) was represented by a square of the respective color. For logging in, subjects had an access code taped to their desk, which could only be used once.

Subjects were told that they would be matched in groups of 35 members who were dispersed throughout the auditorium. Each member of the group had a unique piece of information

---

[9]　The software operated independently of specific platforms using the Internet protocol and was derived from PINGO, a classroom communication software [45], by adding the functionality of sending information to the students.

corresponding to the color of one of the 36 objects from the cardboard box. Subjects were assigned randomly rather than with their immediate neighbors in order to prevent direct communication within the voting group. For practical reasons, the matching of participants into groups was carried out after all decisions were made. One group was selected for payout and each member of this group received € 10 in case that the color with most votes was actually the majority color in the urn.

With 35 group members and 36 objects in the box, one object was left unassigned. This physically conveys the idea that it is unlikely (but possible) that none of the group members might have received the perfect information without having to rely on probabilities.

These rules were explained with a short animated presentation—see online Appendice A–F and Video S1. After the presentation of the rules, we told subjects that we would now proceed under an exam protocol and that anyone caught talking would be excluded.

Before sending out information about colors, we asked subjects several control questions in order to get some indication of their comprehension. We did not provide feedback about correct answers in order to maintain the "natural" heterogeneity. At the end of the round, subjects received information on their mobile device about the voting outcome of their group, whether this outcome was correct, and whether their group was selected for payout. Then, we surprised subjects with the announcement that the voting would be repeated with a new urn and a new draw of 36 objects. A second sealed cardboard box containing the 36 draws from the second urn was presented and placed on the desk next to the box from the first round.

Before starting the second round, we gave subjects 5 min to freely discuss with their neighbors. After the five minutes, subjects were again put under the exam protocol, i.e., no talking or looking around. They then received their signals and cast their votes.

At the very end, we sent subjects a brief questionnaire on their devices in which we inquired about their gender and age. In addition, we wanted to assess whether they were aware of the efficient strategy. We did so by asking subjects to imagine that they would play with robots that could be programmed to follow a certain behavior. We then wanted to know how they would program these robots. They could specify whether robots should vote for blue or green or abstain depending on the information received by the robot.

## 5. Results

The participants in this study were almost 600 students from a large first-year class ("Introduction to Business Administration") at Paderborn University. It was run on two consecutive days and the overwhelming majority of those present participated. Participants were able to "leave" and "enter" the study at any time by disconnecting from or reconnecting to the server. Still, only 3% were lost during the actual experiment, which might be expected for purely technical reasons (network interruptions, low battery). Attrition during the ex-post questionnaire was larger, particularly on the first day, for many participants had to move on to the next lecture. Our analysis will focus on the around 590 participants with imperfect information whose behavior determined whether or not the efficient equilibrium was played. (Numbers for perfectly informed subjects were too low for a meaningful analysis.) The whole experiment took between 20–27 min and average earnings among subjects in groups selected for payout were € 7.50.

### 5.1. Prerequisites

If we want to study whether a group can overcome the problems of unawareness about the efficient strategy and evolve to a more sophisticated use of information, we need to check whether the starting point exhibits such unawareness and whether there is still room for sophistication.

Some evidence for unawareness comes from the control questions. About 6% incorrectly believed that green was more likely to be the majority color when their signal was blue. Around 3% claimed

that voting blue meant that blue became less likely to be the majority color in their group; 30% said that them voting blue would not affect the outcome of the vote.[10]

More importantly, participants did not agree on how to optimally behave. The majority programmed their voting robots to vote regardless of whether they were perfectly or imperfectly informed, which suggests that they are unaware of the efficient only-expert-votes equilibrium. Slightly more than a quarter restricted the robots to voting only when information was perfect and is hence likely to be aware of the efficient equilibrium.The remaining quarter's programming was all over the place—see Table 1. This shows disagreement on how to optimally behave even at the end of the experiment, i.e., after people had the chance to talk to each other. It thus seems likely that these disagreements were more pronounced before subjects communicated.

**Table 1.** Only about a quarter of the subjects programmed robots to act consistently with the efficient equilibrium, i.e., to vote for their color if they are perfectly informed and to abstain otherwise.

|  |  | Day 1 | | | | Day 2 | | | |
|---|---|---|---|---|---|---|---|---|---|
|  |  | Imperfectly Informed | | | | Imperfectly Informed | | | |
|  | **Vote \*** | $-$ | **O** | **+** | $\Sigma$ | $-$ | **O** | **+** | $\Sigma$ |
| **perfectly** | $-$ | 1.2% | 6.7% | 5.5% | 13.5% | 4.5% | 3.0% | 4.8% | 12.3% |
| **informed** | $+$ | 9.2% | 25.8% | 51.5% | 86.5% | 5.5% | 29.1% | 53.0% | 87.7% |
|  | $\Sigma$ | 10.4% | 32.5% | 57.1% | 163 | 10.1% | 32.2% | 57.8% | 398 |

\* $-$ vote against information, O abstain, + vote for information.

Subjects not only expressed disagreement in the un-incentivized question on how to program voting robots; the discrepancy was also reflected in actual behavior. As expected and hoped for, first-round voting was far from the efficient only-expert-votes equilibrium. On both days and in both rounds, almost 80% of non-experts voted for their color—see Table 2. The group was not capable of reaping the efficiency gains from holding back and letting the expert decide. The probability that the majority in a group coincides with the true state in the first round can be computed to be 51.1% (using the actual behavior in the respective formula in Appendix B) and is hence only slightly better than mere guessing. The expert-vote and even the all-vote benchmark of 97.2% and 61.0% were thus missed by a wide margin. The bad performance was also due to a considerable number of imperfectly (and even some perfectly) informed subjects who voted against their signal, which (as we argue in Appendix F) can be traced back to people who misunderstood the game.

Summarizing these observations, initial behavior in our setting exhibits heterogeneity and leaves room for better coordination and sophistication.

**Table 2.** Voting behavior.

| Day | Round | Total | Imperfectly Informed | | | Total | Perfectly Informed | | |
|---|---|---|---|---|---|---|---|---|---|
|  |  |  | + | O | $-$ |  | + | O | $-$ |
| both | 1 | 594 | 78.1% | 6.4% | 15.5% | 16 | 81.3% | 0% | 18.7% |
| both | 2 | 588 | 67.2% | 13.8% | 19.0% | 16 | 100.0% | 0% | 0% |
| 1 | 1 | 199 | 77.4% | 8.5% | 14.1% | 5 | 100.0% | 0% | 0% |
| 1 | 2 | 196 | 68.4% | 20.4% | 11.2% | 5 | 100.0% | 0% | 0% |
| 2 | 1 | 395 | 78.3% | 5.3% | 16.2% | 11 | 72.7% | 0% | 27.3% |
| 2 | 2 | 392 | 66.6% | 15.0% | 18.4% | 11 | 100.0% | 0% | 0% |

+ vote for own information, O abstain, $-$ vote against own information.

---

[10]　This answer might actually be right if subjects (wrongly) believe that pivotal situations never arise.

*5.2. Changes in Voting Behavior*

The best-reply to the actual first-round behavior of only around 6% abstaining was to vote for one's signal—recall Figure 1. On the one hand, one might thus expect the share of votes to rise. On the other hand, abstaining becomes optimal if participants believe that sufficiently many of them will start abstaining after communicating (as limited as this communication may be).

From the first to the second round, the number of abstention increased from 6.4% to 13.8%—see Figure 3. The number of participants who switched to abstention was much higher than those who switched away from abstention, an increase that is highly significant (*p*-value for McNemar test was below 0.001)—although it was much smaller (and not statistically significant) on the first day.

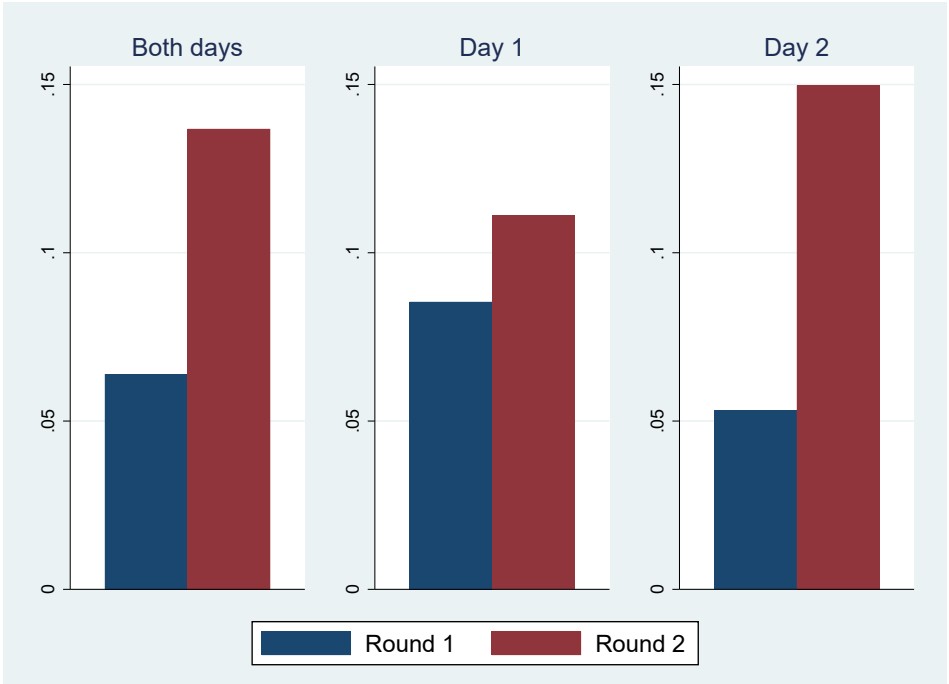

**Figure 3.** The share of abstentions increases.

The increase in abstentions was not large enough to improve performance. Holding the shares of participants who voted against their signal constant, the probability that the majority in a group voted for the true state computed from the observed behavior slightly dropped to 51.0% (this was true for both days). The increase simply did not push abstentions to a level at which gains could be expected—recall Figure 2. If participants started abstaining in the hopes that sufficiently many others would also do so, this hope was not fulfilled.

**Result 1.** *Subjects are more likely to abstain in the second round, although this was not the best response to actual behavior.*

*5.3. Explanations for the Switch to Abstention*

There are three reasons why people might switch from voting to abstaining between the first and second round. First, they learn the efficient strategy during the discussion. Second, they already know about this strategy but do not dare to act on it because they are afraid that others might be voting. The group discussion could then reduce the fear in so far as it becomes clear that others also know the strategy and are willing to act on it. Third, they discover the strategy by pondering the situation without the help of others during the discussion time. This may seem unlikely because apart from the discussion, there is no helpful new information from which to learn. Still, it cannot be excluded.

If the increase is related to learning about the only-expert votes equilibrium between the rounds either individually or from others, the increase should be associated with knowledge about the efficient strategy. This seems to be the case. People who later programmed their robot to that equilibrium were more likely to have switched to abstention (86%) than away from it (14%). On the other hand, people who did not program their robots to the equilibrium were more likely to have switched away from abstention (66%) than to it (33%)—see Table 3. The difference between the two groups is significant at any conventional level. That switchers are more likely to later know about the equilibrium suggests that they learned about it between the rounds.

**Table 3.** Programming the efficient equilibrium is associated with starting to abstain.

| Programmed Efficient Eq. | Start Abstaining | | |
| --- | --- | --- | --- |
| | No | Yes | Σ |
| no | 66.7% | 33.3% | 12 |
| yes | 14.0% | 86.0% | 50 |
| Σ | 15 | 47 | 62 |

Fisher's exact test is significant at any conventional level.

We can distinguish more systematically between the reasons by examining how the number of abstaining neighbors in the first round affects abstention in the second round. If participants already know about the strategy but start believing that other members will act on it, their individual beliefs should increase the number of neighbors who abstain around them. With respect to individual learning between the rounds, one would a priori perhaps expect that there is no correlation with the number of neighbors who abstained. If people sit together with friends whose abilities are correlated with their own, however, one would also observe a spurious correlation between the number of abstaining neighbors (indicating a higher ability in this local area) and the likelihood to start abstaining in the second round. If participants learn the efficient strategy from their neighbor, one abstaining neighbor who is willing to share the strategy suffices. The last two channels assume that abstention in the first round is an indicator for knowing about the efficient strategy. This assumption is plausible since the two are highly correlated—see Appendix E.

In order to find out about the relationship between switches to abstention and the number of neighbors who abstained before, we regress the change to abstention after either voting in line with or against one's own signal on dummies indicating whether one, two, or many neighbors abstained in the previous round. In our preferred specification (G2SLS), we allow for local correlation in the dependent variable as well as in the error term.[11] We only used the 525 subjects who did not abstain in the first round.

The presence of one neighbor who abstained in the first round was highly significantly correlated with abstention in the next round—see Table 4. With such a neighbor, it was 13% more likely that an individual changed from voting to abstaining in the next round.

---

[11] More precisely, we estimate a spatial autoregressive model with autocorrelated errors (SARAR) using a generalized spatial two-stage least squares regression (G2SLS). As a robustness check, we use maximum likelihood (ML) estimates and a plain vanilla regression without local effects (OLS). Unsurprisingly, abstention in the second round is locally correlated: individuals are significantly more likely to abstain when their neighbors abstain (see $\lambda$ in the table). The correlation $\rho$ between neighbors' errors, i.e., after taking the effect of observables on abstention into account, is negative and significant but only using G2SLS.

**Table 4.** Regression for switches to abstentions between the rounds.

| | G2SLS | ML | OLS |
|---|---|---|---|
| Neighbors abstaining in round 1... | | | |
| ...at least one | 0.134 *** | 0.155 *** | 0.165 *** |
| | (0.051) | (0.044) | (0.047) |
| ...two | −0.055 | −0.034 | −0.026 |
| | (0.092) | (0.075) | (0.077) |
| ...more than two | 0.053 | 0.072 | 0.080 |
| | (0.257) | (0.177) | (0.181) |
| Day 1 | −0.016 | −0.016 | −0.016 |
| | (0.025) | (0.030) | (0.031) |
| Neighbor abstained × Day 1 | −0.156 * | −0.118 * | −0.116 |
| | (0.083) | (0.069) | (0.073) |
| Neighbor expert in Round 1 | −0.002 | 0.004 | 0.004 |
| | (0.032) | (0.038) | (0.042) |
| Intercept | 0.024 | 0.058 *** | 0.087 *** |
| | (0.020) | (0.022) | (0.018) |
| local correlation in abstention $\lambda$ | 0.096 *** | 0.042 * | |
| (round 2) | (0.031) | (0.023) | |
| local correlation in error terms $\rho$ | −0.084 *** | −0.038 | |
| (round 2) | (0.030) | (0.030) | |
| Number of observations | 525 | 525 | 525 |

Significance levels: *10%, ** 5%, ***1%.

**Result 2.** *Having a neighbor who previously abstained is associated with a higher probability of abstention. Having several abstaining neighbors is not associated with a further increase in the likelihood to abstain.*

This result is consistent with learning the sophisticated strategy from a neighbor but not with learning independently or learning to trust that others abstain.

*5.4. Reasons for the Difference between the Days*

There was a marked difference between both days. Participants were much less likely to switch to abstention on the first than on the second day—recall Figure 3.

Attributing this difference to a specific feature is difficult because the two days differed in many ways, including starting time (whether a lecture is at Monday 7:30 or Tuesday 9:15 is important to many students, leading to potential selection effects) and study programs to which students signed up. There were also substantially fewer students on the first day, which in principle can facilitate coordination. On the other hand, students were less densely seated on the first day: only about a third of seats were filled and participants were widely dispersed throughout the auditorium. Some did not even have the opportunity to discuss with their neighbors because they had no neighbors—see Figure 4. Information could thus travel less well in the room on this day.

While we cannot identify which of these confounding factors caused the difference, there is a hint. Examining the regression, we see that switches to abstention are not related to the day in itself. The coefficient of the day dummy is small and insignificant—see Table 4. This suggests that we are not dealing with a direct effect of the day. Instead, abstention seemed to be less contagious on the first day. While on the second day, having an abstaining neighbor was associated with an increase of abstention of 13%, the increase was 15 percentage points lower on the first day; the respective coefficient of the interaction term of whether one neighbor abstained and the day 1 dummy was statistically significant (albeit at the 10% level). This is consistent with people not bothering to explain the sophisticated strategy to their neighbors when information cannot travel well.

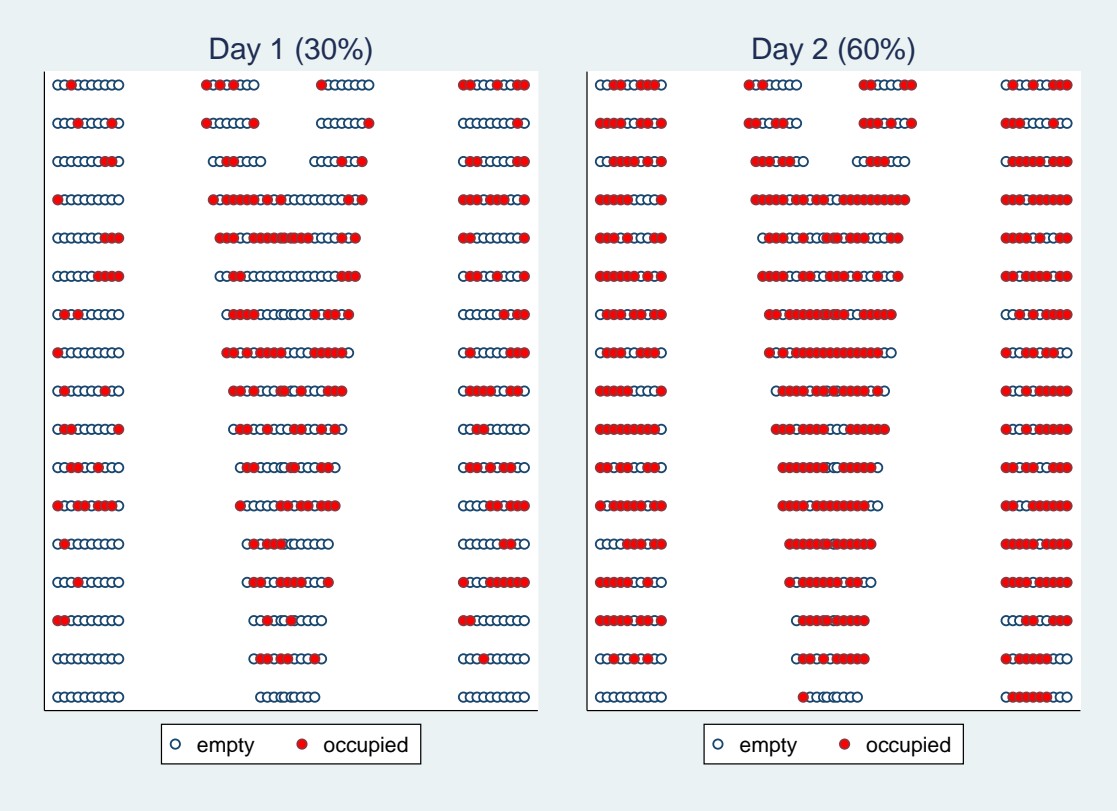

**Figure 4.** Density of seating on both days.

## 6. Conclusions

Organisations, states, or companies can be in turmoil if they lack a functioning culture or understanding of how to act appropriately, i.e., a system of self-enforcing rules. The reason is diverging views of the facts and what to make of them and no central entity to guide the group. Some rule systems are more efficient but require sophisticated reasoning and trust, whereas others are more resilient in that they are robust to errors and confusion. Given the chance to talk but only locally, is there any chance that the more efficient system might emerge? Or will the organization take the path toward the resilient but less efficient equilibrium, which does not require a common understanding, trust, and communication?

Here, we studied this question using a voting game in a setting that included not only diverging views of the facts but also naturally occurring differences in what to make of them: a large first-year undergraduate class. The situation was not favorable for the efficient but sophisticated equilibrium to arise in. The initial behavior was already very close to the more resilient but less efficient equilibrium. Surprisingly, we found that after the opportunity to communicate, the group started moving in the direction of the more sophisticated equilibrium.

Using econometric analysis, we tentatively concluded that the increase came from students teaching each other about the more sophisticated strategy—even though this strategy was not the best response to what they experienced. This is a sign that it is possible for a group without central agency to move in a promising direction.

**Supplementary Materials:** The following are available online at http://www.mdpi.com/2073-4336/11/3/31/s1, Video S1: Experimental Instructions as Animated Slides.

**Author Contributions:** Conzeptualization, W.S. and R.V.; experimental design, W.S. and R.V.; software supervision, D.S.; formal analysis, W.S. and R.V.; data collection, D.S.; Implementation of design, D.S. and W.S.; writing, W.S. and R.V. All authors have read and agreed to the published version of the manuscript.

**Funding:** This research received no external funding.

**Acknowledgments:** This article benefited from comments by Jean Tyran, Ben Greiner, Markus Reisinger, Johannes Münster and Clemens Puppe as well as from participants at research seminars in Dortmund, Heidelberg, and Århus. All errors remain our own. We especially want to thank Jürgen Neumann for his excellent programming work.

**Conflicts of Interest:** The authors declare no conflict of interest.

## Appendix A. Proofs for the Analysis of the Stage Game

**Lemma A1** (Strict dominance). *Under the strategic uncertainty assumption, voting for the signal, $q_i(s_i, 0) = 1$, strictly dominates voting against it, $r_i(s_i, 0) = 1$.*

**Proof.** Individual $i$ can only alter the outcome in two cases: (i) others' votes are tied or (ii) there is a single-vote majority (among the other group members) in favor of the true state.[12] In both cases, voting for her signal yields a payoff of one if this signal matches the true state of the world, which happens with probability $P(s_i = \theta)$, or if she votes against her signal and the signal does not match the true state, which happens with probability $P(s_i \neq \theta)$. Since $P(s_i = \theta) > P(s_i \neq \theta)$, the former happens more often and voting in line with one's own signal yields a higher expected payoff in both case (i) and (ii). In all non-pivotal cases, the individual is indifferent. For a perfectly informed voter, the analysis is exactly the same.[13] For strict dominance, observe that both cases, (i) and (ii), occur with positive probability under the strategic uncertainty assumption. □

**Lemma A2** (Dominant strategy when perfectly informed). *Under the strategic uncertainty assumption, voting strictly dominates abstaining for perfectly informed individuals ($t_i = 1$).*

**Proof.** Recall that the outcome is only affected in case of a one-vote majority for the true state or a tie. Voting and abstaining both lead to a payoff of one in case of a one-vote majority for the true state. In case of a tie, voting shifts the payoff from zero to one because the vote is in line with the true state with certainty, while the outcome after abstaining remains zero. Since ties occur with positive probability, the dominance is thus strict. □

**Proof for Proposition 1.** For the proof use the best response function of an imperfectly informed participant from Figure 1. Firstly, if all players vote their signal, we are at the right end of the figure and there are gains for player $i$ to also vote for her signal. (This is also true for the player with perfect information.) Secondly, if only the perfectly informed player votes for her signal and all others abstain (only-expert-votes equilibrium), we are at the left end of the figure and there are substantial losses if player $i$ votes. If the perfectly informed participant would not vote in this equilibrium, this would lead to a tie and reduce the probability of obtaining a positive payoff to zero. Thirdly, if a fraction of around 70% of the imperfectly informed players as well as the perfectly informed player are voting for their signal while the others abstain, all imperfectly informed players would be indifferent between voting for their signal or abstaining, while the perfectly informed player gains from voting her signal. There are, of course, other equilibria of the game. These, however, require that individual players are unable to affect the outcome and thus clash with our strategic uncertainty assumption. □

**Proof for Proposition 2.** If only the perfectly informed member votes, the majority chooses the true state whenever this member is present in the group. This happens in $n - 1 = 35$ of $n = 36$ cases. The formulas used for the calculation of the expected payoff under more general conditions and in

---

[12] If there is a single-vote majority favoring the wrong state, then $i$'s vote can either boost the majority or convert it into a tie. In either case, the payoff is one and the same, namely, zero. If the payoff in case of a tie had been determined with a flip of a fair coin, this lemma would still go through.

[13] Note, the only difference between a perfectly and imperfectly informed voter is the strength of the signal. In the imperfect case the signal is only informative $1/2 < P(s_i = \theta) < 1$; in the perfect case, it is truth revealing $P(s_i = \theta) = 1$. The argument above only relies on $P(s_i = \theta) > 1/2$ and so applies to both types.

particular when everyone votes can be found in Appendix B. The mixed-strategy equilibrium has neither the benefit of fully exploiting the perfect information nor fully aggregating the imperfect signals—see Figure 2. □

## Appendix B. Expected Payoff When $q$ Vote for and $r$ against Their Draw

**Lemma A3.** *$q_0$ non-experts vote in line with their draws and $r_0$ against them, and the rest abstain; $q_1$ experts vote for their signals and $r_1$ against the. The probability that the majority votes for the correct state is:*

$$
\begin{aligned}
P(v > 0) = {} & \frac{1}{36} \sum_{k=0}^{35} \sum_{l=0}^{35-k} Q_0(k,l) P_0(k,l) \\
& + \frac{35}{36} q_1 \sum_{k=0}^{34} \sum_{l=0}^{34-k} Q_1(k,l) P_1^+(k,l) \\
& + \frac{35}{36} r_1 \sum_{k=0}^{34} \sum_{l=0}^{34-k} Q_1(k,l) P_1^-(k,l) \\
& + \frac{35}{36} (1 - q_1 - r_1) \sum_{k=0}^{34} \sum_{l=0}^{34-k} Q_1(k,l) P_0(k,l),
\end{aligned}
\tag{A1}
$$

*where*

$$
\begin{aligned}
Q_0(k,l) &:= \frac{35!}{k!l!(35-k-l)!} q_0^k r_0^l (1 - q_0 - r_0)^{35-k-l} \\
Q_1(k,l) &:= \frac{34!}{k!l!(34-k-l)!} q_0^k r_0^l (1 - q_0 - r_0)^{34-k-l}.
\end{aligned}
\tag{A2}
$$

*and*

$$
\begin{aligned}
P_0(k,l) &:= \sum_{x=0}^{k} \sum_{y>\max\{0,\frac{k+l}{2}-x\}}^{l} \frac{\binom{M}{x+l-y}\binom{N-M}{y+k-x}}{\binom{N}{k+l}} \cdot \frac{\binom{k}{x}\binom{l}{l-y}}{\binom{k+l}{x+l-y}}. \\
P_1^+(k,l) &:= \sum_{x=0}^{k} \sum_{y\geq\max\{0,\frac{k+l}{2}-x\}}^{l} \frac{\binom{M}{x+l-y}\binom{N-M}{y+k-x}}{\binom{N}{k+l}} \cdot \frac{\binom{k}{x}\binom{l}{l-y}}{\binom{k+l}{x+l-y}}. \\
P_1^-(k,l) &:= \sum_{x=0}^{k} \sum_{y>\max\{0,\frac{k+l}{2}+1-x\}}^{l} \frac{\binom{M}{x+l-y}\binom{N-M}{y+k-x}}{\binom{N}{k+l}} \cdot \frac{\binom{k}{x}\binom{l}{l-y}}{\binom{k+l}{x+l-y}}.
\end{aligned}
\tag{A3}
$$

**Proof.** If no expert is in the sample, the probability of $q_0$ who vote in line and $r_0$ who vote against their signal, result in a sample of $k$ voting for their signal, $l$ against it, while $35 - k - l$ abstain. The probability for this event is:

$$
Q_0(k,l) := \frac{35!}{k!l!(35-k-l)!} q_0^k r_0^l (1 - q_0 - r_0)^{35-k-l}.
$$

Say that $x$ of those who vote in line with their draw receive the true signal, while $y$ of those who vote against their signal receive the wrong signal. Then, the true state receives $x + y$ votes and a majority for the true state results if and only if $x + y > \frac{k+l}{2}$. This requires that $x + l - y$ of the $M$ balls of the majority color to be drawn and $y + k - x$ of the $N - M$ balls of the minority color. Moreover, of the total of $k + l - y$ true signals that are drawn, $x$ must come from the $k$ who vote for and $l - y$ from the $l$ who vote against their signal. The probability for such a specific combination of $(x, y, k, l)$ is thus:

$$
\frac{\binom{M}{x+l-y}\binom{N-M}{y+k-x}}{\binom{N}{k+l}} \cdot \frac{\binom{k}{x}\binom{l}{l-y}}{\binom{k+l}{x+l-y}}.
$$

Summing these probabilities for all cases where $x + y > \frac{k+l}{2}$ leads to:

$$P_0(k,l) := \sum_{x=0}^{k} \sum_{y > \max\{0, \frac{k+l}{2}-x\}}^{l} \frac{\binom{M}{x+l-y}\binom{N-M}{y+k-x}}{\binom{N}{k+l}} \cdot \frac{\binom{k}{x}\binom{l}{l-y}}{\binom{k+l}{x+l-y}}. \tag{A4}$$

If an expert is in the sample, the 34 players consist of $k$ individuals who vote in line with and $l$ who vote against their signal, while $34 - k - l$ abstain with probability:

$$Q_1(k,l) := \frac{34!}{k!l!(34-k-l)!} q_0^k r_0^l (1 - q_0 - r_0)^{34-k-l}.$$

In $q_1$ percent of the cases, the expert votes in line with her signal and a tie among the remaining 34 members is now enough for a majority for the true state, leading to the following version of (A4):

$$P_1^+(k,l) := \sum_{x=0}^{k} \sum_{y \geq \max\{0, \frac{k+l}{2}-x\}}^{l} \frac{\binom{M}{x+l-y}\binom{N-M}{y+k-x}}{\binom{N}{k+l}} \cdot \frac{\binom{k}{x}\binom{l}{l-y}}{\binom{k+l}{x+l-y}}. \tag{A5}$$

In $r_1$ percent of the cases, the expert votes against her signal. A majority for the true state thus requires at least a two-vote lead among non-experts: $x + y \geq \frac{k+l}{2} + 2$ and the probability becomes:

$$P_1^-(k,l) := \sum_{x=0}^{k} \sum_{y > \max\{0, \frac{k+l}{2}+1-x\}}^{l} \frac{\binom{M}{x+l-y}\binom{N-M}{y+k-x}}{\binom{N}{k+l}} \cdot \frac{\binom{k}{x}\binom{l}{l-y}}{\binom{k+l}{x+l-y}}. \tag{A6}$$

In case that the expert abstains, we are back to the same probability as if no expert would be in the sample, with the difference that there is now one vote less. Putting together the four cases, we get the result. $\square$

**Appendix C. Individual Gains from Voting When $q$ Vote for and $r$ against Their Draw**

**Lemma A4.** *Suppose there is no expert among the members of the group and an even number of its members $k + l$ vote, where $k$ vote in line and $l$ against their signals. Then, the expected gain of a non-expert from voting relative to abstaining for $k > l$ is:*

$$G_E^{NE}(k,l) := \frac{M}{N} \cdot \sum_{i=0}^{l} \frac{\binom{M-1}{\frac{k+l}{2}+l-2i}\binom{N-M}{k-\frac{k+l}{2}+2i}}{\binom{N-1}{k+l}} \frac{\binom{\frac{k+l}{2}+l-2i}{l-i}\binom{k-\frac{k+l}{2}+2i}{i}}{\binom{k+l}{l}}$$

*Additionally, for $k < l$ it is:*

$$G_E^{NE}(k,l) := \frac{M}{N} \cdot \sum_{i=0}^{k} \frac{\binom{M-1}{l-\frac{k+l}{2}+2i}\binom{N-M}{k+\frac{k+l}{2}-2i}}{\binom{N-1}{k+l}} \frac{\binom{l-\frac{k+l}{2}+2i}{i}\binom{k+\frac{k+l}{2}-2i}{k-i}}{\binom{k+l}{k}}$$

**Proof.** Suppose a non-expert votes for her color. This generates a gain of one if and only if (i) the true state is actually that of her color and (ii) the other non-experts' votes are tied. The probability of the signal matching the true state is: $\frac{M}{N}$. The votes of the other $k + l$ non-experts are tied whenever the number of voters who vote for their signal with the correct signal is equal to the number of voters who vote against their signal and also have the correct signal.

If there are more people who vote for their signal, $k > l$, the median voter is in this group and we get the following cases. First, all $\frac{k+l}{2}$ votes for the correct state may come from this group, the rest of this group $k - \frac{k+l}{2}$ then must have signals indicating the wrong state and all $l$ from the counter vote-group must have the correct signal. Accordingly, from the $\frac{k+l}{2} + l$ correct draws, $l$ must have landed in the counter group. Then, $l + k - \frac{k+l}{2} = \frac{k+l}{2}$ vote against and there is a tie. There is also a tie

if one voter voting his signal and one voting against his signal receive a wrong instead of a correct signal. This can be continued until all in the counter group receive the correct signal.

If there are less people who vote for their signal, $l < k$, the median voter is in the counter group and we get the following cases. First, if all $\frac{k+l}{2}$ votes from the counter group have the wrong signal, the rest of this group $l - \frac{k+l}{2}$ then must have signals indicating the correct state and all $k$ from the pro-group the wrong signal. (The event must be weighed with the probability that all $k$ indeed have the wrong signal.) Then, $k + l - \frac{k+l}{2} = \frac{k+l}{2}$ vote against and there is a tie. There is also a tie if one voter voting his signal and one voting against his signal receive a correct instead of the wrong signal. This can be continued until all in the pro group receive the correct signal.　□

**Lemma A5.** *Suppose there is no expert among the members of the group and an odd number $k + l$ of non-experts who vote, where $k$ is those who vote for and $l$ those who vote against their signal. Then, the expected gain of a non-expert from voting relative to abstaining for $k > l$ is:*

$$G_O^{NE}(k,l) := -\frac{N-M}{N} \cdot \sum_{i=0}^{l} \frac{\binom{M}{\frac{k+l+1}{2}+l-2i}\binom{N-M-1}{k-\frac{k+l+1}{2}+2i}}{\binom{N-1}{k+l}} \frac{\binom{\frac{k+l+1}{2}+l-2i}{l-i}\binom{k-\frac{k+l+1}{2}+2i}{i}}{\binom{k+l}{l}}$$

*And for $k < l$ it is:*

$$G_O^{NE}(k,l) := -\frac{N-M}{N} \cdot \sum_{i=0}^{k} \frac{\binom{M}{l-\frac{k+l+1}{2}+2i}\binom{N-M-1}{\frac{k+l+1}{2}+k-2i}}{\binom{N-1}{k+l}} \frac{\binom{l-\frac{k+l+1}{2}+2i}{i}\binom{\frac{k+l+1}{2}+k-2i}{k-i}}{\binom{l+k}{k}}$$

**Proof.** Suppose a non-expert votes her color. This generates a loss of one if and only if (i) her own color does not match the true state and (ii) the other non-experts favor the true state by a single-vote majority. The probability that a signal is the wrong color is: $\frac{N-M}{N}$. The votes of the other $k + l$ non-experts have a single vote majority whenever the number of voters who vote for their signal with the correct signal is one larger than the number of voters who vote against their signal and also have the correct signal.

If $k > l$, all votes for the true state may come from $\frac{k+l+1}{2}$ of the $k$ who vote their signal, while the remainder of the group, $k - \frac{k+l+1}{2}$ has the signal of the wrong state and votes against. Together with $l$ from the group who vote against their correct signal, we get a lead of one vote for the true state: $k - \frac{k+l+1}{2} + l = \frac{k+l+1}{2} - 1$. For this, all of the $l + \frac{k+l+1}{2}$ correct draws must end up in the counter-vote group. Another one-vote lead can be found if there is one less correct draw in both groups. The number of correct draws can be reduced until no counter voter has a correct draw anymore.

If $k < l$, all votes for the true state may come from $\frac{k+l+1}{2}$ of the $l$ who vote against their wrong signal, while the remainder of the group, $l - \frac{k+l+1}{2}$ has the correct signal and votes against it. Adding the $k$ voters who vote for their wrong signal, we get a one-vote lead.For this $k$ of the $frac{k+l+1}{2}+k$ wrong signals must have ended up in the group voting their signal. Another one-vote lead can be found if there is one more correct draw in both groups. The number of correct draws can be increased until all of those voting for their draw have a correct draw.　□

**Lemma A6.** *Suppose there is an expert among the members of the group who votes for her signal and an even number $k + l$ of non-experts vote, where $k$ is those who vote for and $l$ those who vote against their signal. Then, the expected gain of a non-expert from voting relative to abstaining for $k > l$ is:*

$$G_E^{WE}(k,l) := -\frac{N-M}{N} \cdot \sum_{i=0}^{l} \frac{\binom{M}{\frac{k+l}{2}+l-2i}\binom{N-M-1}{k-\frac{k+l}{2}+2i}}{\binom{N-1}{k+l}} \frac{\binom{\frac{k+l}{2}+l-2i}{l-i}\binom{k-\frac{k+l}{2}+2i}{i}}{\binom{k+l}{l}}$$

*And for k < l it is:*

$$G_E^{WE}(k,l) := -\frac{N-M}{N} \cdot \sum_{i=0}^{k} \frac{\binom{M}{l-\frac{k+l}{2}+2i}\binom{N-M-1}{k+\frac{k+l}{2}-2i}}{\binom{N-1}{k+l}} \frac{\binom{l-\frac{k+l}{2}+2i}{i}\binom{k+\frac{k+l}{2}-2i}{k-i}}{\binom{k+l}{k}}$$

**Proof.** Suppose a non-expert votes for her color. This generates a loss of one if and only if (i) the true state is actually not the color of her signal and (ii) non-experts' votes are tied. (In this case the expert decides and chooses the true state). The probability the signal is the wrong color is: $\frac{N-M}{N}$. The calculation when other non-experts are tied, is essentially the same as in Lemma A4. The only difference is that this time, the voter has drawn the wrong color. □

**Lemma A7.** *Suppose there is an expert among the members of the group who votes for her signal and an odd number $k + l$ of non-experts who vote, where k is those who vote for and l those who vote against their signal. Then, the expected gain of a non-expert from voting relative to abstaining for $k > l$ is:*

$$G_O^{WE}(k,l) := \frac{M}{N} \cdot \sum_{i=0}^{l} \frac{\binom{M-1}{k-\frac{k+l+1}{2}+2i}\binom{N-M}{\frac{k+l+1}{2}+l-2i}}{\binom{N-1}{k+l}} \frac{\binom{k-\frac{k+l+1}{2}+2i}{i}\binom{\frac{k+l+1}{2}+l-2i}{l-i}}{\binom{k+l}{l}}$$

*And for k < l it is:*

$$G_O^{WE}(k,l) := \frac{M}{N} \cdot \sum_{i=0}^{k} \frac{\binom{M-1}{\frac{k+l+1}{2}+k-2i}\binom{N-M}{l-\frac{k+l+1}{2}+2i}}{\binom{N-1}{k+l}} \frac{\binom{\frac{k+l+1}{2}+k-2i}{k-i}\binom{l-\frac{k+l+1}{2}+2i}{i}}{\binom{l+k}{k}}$$

**Proof.** Suppose a non-expert votes her color. This generates a gain of one if and only if (i) the true state matches her color and (ii) the other non-experts' votes favor the wrong state by a single-vote margin. (The expert's vote then leads to a tie). The probability the signal matches the true state is: $\frac{M}{N}$. The probability of the other non-experts favoring the wrong state by a single-vote majority is essentially the same as in Lemma A5 with the difference that now the false state is favored by one vote. □

**Lemma A8.** *If q is the probability that a non-expert votes for her signal and r the probability that she votes against it. Then, the expected gain of a non-expert from voting relative to abstaining is:*

$$\frac{1}{n-1}\underbrace{\sum_{k=0}^{34}\sum_{l=0}^{34-k}\frac{34!}{k!l!(34-k-l)!}q^k r^l (1-q-r)^{34-k-l} \cdot \begin{cases} G_E^{NE}(k,l) & k+l \text{ even.} \\ G_O^{NE}(k,l) & k+l \text{ odd.} \end{cases}}_{=:G^{NE}}$$

$$+\frac{n-2}{n-1}\underbrace{\sum_{k=0}^{33}\sum_{l=0}^{33-k}\frac{33!}{k!l!(33-k-l)!}q^k r^l (1-q-r)^{33-k-l} \cdot \begin{cases} G_E^{WE}(k,l) & k+l \text{ even.} \\ G_O^{WE}(k,l) & k+l \text{ odd.} \end{cases}}_{=:G^E}$$

**Proof.** Consider a non-expert. The probability that an expert has not been drawn (from the remaining $n-1$ candidates) to be a part of the group is $\frac{\binom{1}{0}\binom{n-2}{n-2}}{\binom{n-1}{n-2}} = \frac{1}{n-1}$. In this case, there are $n-1$ other non-experts and the probability that $k$ of them vote for and $l$ against their signal is given by the multinomial in the top line of the expression. With the complementary probability, $\frac{n-2}{n-1}$, there is an expert. In this case, there are $n-2$ other non-experts and the probability that $k$ of them vote for and $l$ against their signal is given by the multinomial expression in the bottom line of the expression. The respective gains $G$ were computed in the preceding lemmas. □

## Appendix D. Session Protocol

Before the lecture, students were instructed by email to bring an Internet capable device such as a laptop or mobile phone. The email also included instructions on how to connect with the wireless network of the university.

The class was taught by one of the authors,[14] focuses on how management decisions might be biased (based on system 1 rather than system 2) in order to motivate that economic theory and econometrics try to overcome those biases; it features a prisoner's dilemma, introduces the notion of externality and brings a "soft" re-statement of the first fundamental theorem of welfare. It did not cover voting theory or any other aspects of game theory beyond the prisoner's dilemma.

The study was run on the 9th and 10th of November 2015 in a room that seats 620 people—see Figure A1. Originally, the study was scheduled two weeks earlier (the 26th and 27th of October 2015). On the 26th of October, the server broke down when subjects were trying to login. The same happened on the October 27th, although several servers and a load distributor were used. The problem was eventually solved by hiring fast server capacity. Most subjects thus were exposed to the instructions twice. There is no evidence of any intermediary discussions between students in between. Notice that any such discussion would have weakened our finding that more abstention occurs after communicating in the lecture.

On the first day (Monday), the lecture took place at 7:30 and was addressed to students of International Business Studies and Economics & Engineering. Students were asked to stay for the study after the lecture at around 8:40. The number of logged in participants fluctuated by 4 people around 203. This was the overwhelming majority of those present at the lecture. It dropped to about 120 during the ex-post questionnaire, i.e., when the actual study was over at around 9:00 and when it was time to go to the next class.

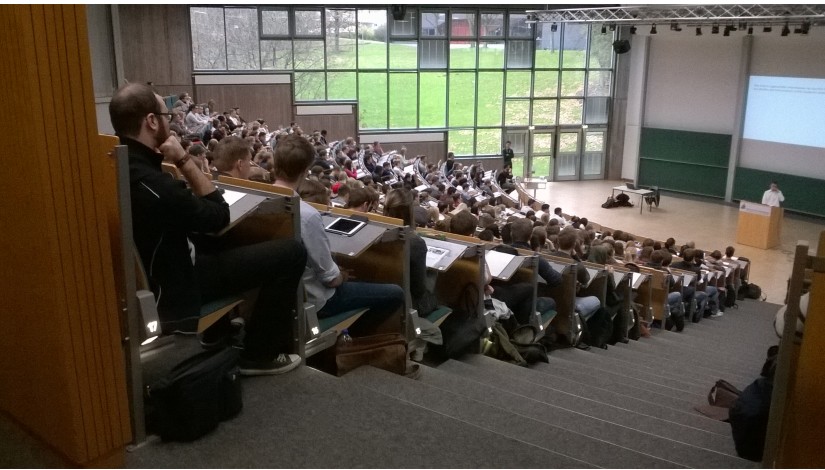

**Figure A1.** Participants listening to the presentation of the instructions.

On the second day (Tuesday), the lecture started at 9:15 and was aimed at students with a major in Economics. The study took place before the lecture and the number of logged in participants fluctuated by 16 people around 400. Again, this was the overwhelming majority of students present. Apart from the drop in ex-post questionnaire answers on the first day, there is no noticeable attrition.

---

14 In principle, there is a danger that students want to show off in front of their teacher by choosing the clever "abstain" strategy. The sheer size of the class, however, renders it impossible for the lecturer to remember the names or faces of anyone and signaling cleverness is thus of little value. If anything, there is more reason for showing off in the first than in the second round; such effects thus run against our finding.

The study lasted 20 min on the first day and seven minutes longer on the second day. The largest chunk of the time was spent on the instructions (6 min). The shortest part was the actual voting which took only 2 min on the first day and 4 min on the second day.

Given the number of participants, the probability of being selected for payout was around $\frac{35}{200} = 17\%$ in each of the two rounds of the first day and $\frac{35}{400} = 8.75\%$ on the second day.

## Appendix E. Abstentions in the First Round Stongly Associated with Programming of Efficient Strategy

Abstention in the first round is an indication of knowing about the efficient only-expert equilibrium. Participants who program their robots in accordance with the efficient strategy are much more likely to abstain—see Table A1. Among those who vote only 23–26% program the efficient strategy, whereas the share is between 56% to 75% among those who abstain. The positive relationship between programming the efficient strategy and abstaining is highly significant on both days.

**Table A1.** Programming the sophisticated strategy is associated with abstentions (in Round 1).

| Day 1 Round 1 | | | | Day 2 Round 1 | | | |
|---|---|---|---|---|---|---|---|
| **Programmed Efficient Eq.** | **Abstaining** | | Σ | **Programmed Efficient Eq.** | **Abstaining** | | Σ |
| | **No** | **Yes** | | | **No** | **Yes** | |
| no | 76.8% | 43.8% | 73.4% | no | 74.0% | 25.0% | 71.4% |
| yes | 23.2% | 56.3% | 26.6% | yes | 26.0% | 75.0% | 28.6% |
| Σ | 116 | 42 | 158 | Σ | 365 | 20 | 385 |

Fisher's exact test is highly significant on both days.

## Appendix F. Voting against One's Information

The data presented in Table 2 indicate that a non-negligable number of participants vote against their own signal. Given our observation that this strategy is dominated, this behavior is unexpected.[15] This section inquires into possible reasons. The analysis here is retrospective. It is an exploratory exercise which means the results should be taken with a grain of salt.

One reason for voting against one's own signal might be cheating. We can distinguish between two types of cheating. First, participants may be sitting next to an expert and see the color of the majority of balls. If they have a different color, it would be optimal for them to vote against their signal. This type of cheating relies on having a sufficiently good view on the neighbors' phone to distinguish between the shape being a circle or a square and identify the color. Second, participants may not be seeing the exact shape of the information but only get an impression of whether there is a dominant color in their local neighborhood. Again, they might vote for this color rather than their own color. While there are a few isolated cases consistent with the first kind of cheating, there is no evidence consistent with the second kind of cheating—see Figure A2.

For a more systematic analysis on whether participants cheat by copying an expert, we run a regression that includes a variable whether an expert with opposing signal is nearby. We control for the second explanation by including a variable that measures the percentage of visible neighbors with an opposing color. Moreover, cheating should be easier if people sit further at the back or closer together, which is why we include the row number and a dummy variable for the first day.

---

[15] Voting against one's own signal seems to be a more widespread phenomenon that is also observed, e.g., by Bouton et al. [46] or Herrera et al. [47].

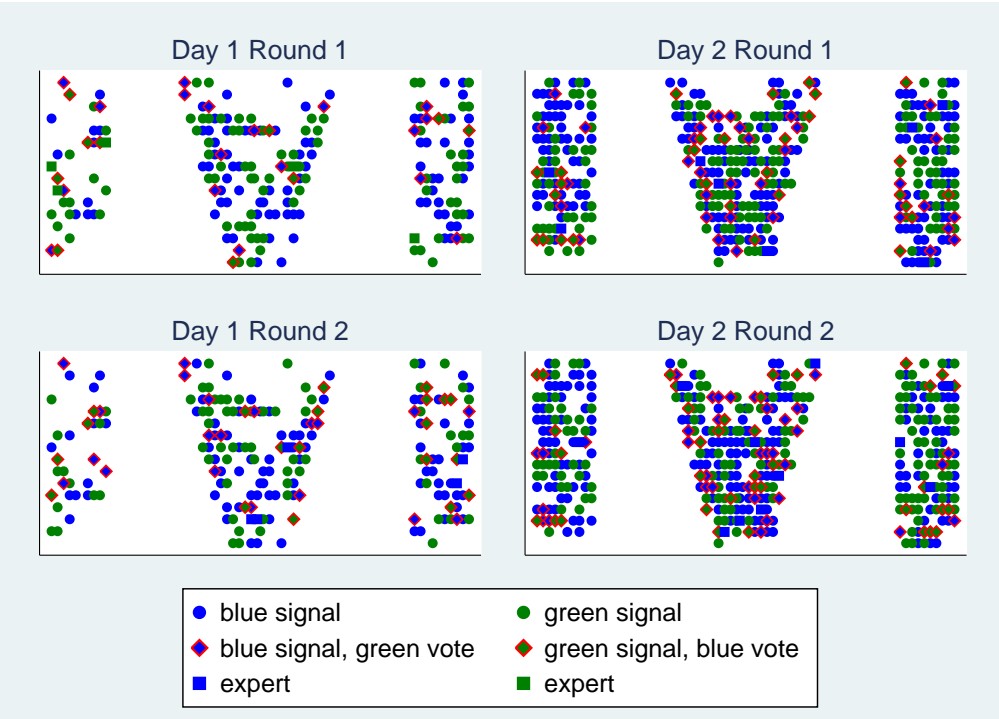

**Figure A2.** Local distribution of votes against own signal.

Before we discuss the results, let us outline two other possible explanations for voting against one's own information. One of them is that some subjects may have been confused or not paid attention. In this case, we speculate that participants might simply vote for whichever choice is presented to them first on the decision screen. In our case, this is the color "green." According to this explanation, participants with blue signals would be more likely to vote against their signal (perhaps less so for the blue colored experts).

Lastly, subjects may have suffered from some misconceptions regarding the nature of the game. We can learn about these from the control questions. About 6% incorrectly believe that green is more likely to be the majority color when their signal was blue. Around 3% claim that voting blue means that blue becomes less likely to be the majority color in their group. 30% say that them voting blue will not affect the outcome of the vote. (This answer might actually not be wrong if subjects interpreted the question to mean that they are unlikely to be pivotal.) For all three cases, we include respective indicator variables.

We ran an ordinary least squares regression in which we explain first round behavior using the above control variables. We took the first round because participants have not yet talked to each other. (When allowing for spatial autocorrelation in the dependent variable and the error term or using both rounds, results remain essentially the same.[16])

The proxy variables indicating cheating are not significant and in the case of opposite expert information even have the wrong sign (having an expert with opposing information renders it less likely that I vote against my signal)—see Table A2. The key variables are related to misconceptions about the nature of the game. Participants who believe that having received a green signal goes along with a higher likelihood of the majority of balls in the urn being blue are 30% more likely to vote against their signal. Similarly, participants who believe that voting blue renders it less likely to be the elected color are 27% more likely to vote against their signal.

---

[16] When estimating both rounds with an OLS (errors clustered at individual level) being expert reduces contrarian votes at the 10% level.

Votes against one's signal thus seem to be driven by misconceptions about the nature of the game. This finding is in line with [7]. Just as on any exam, it is not uncommon to see some students misunderstand a question. In our case, having some fraction of participants misunderstand parts is a natural feature that makes it even less likely that participants abstain initially but gives them a chance to learn during communication. A sizable share of subjects stops voting against their information in the second round (44% on day 1 and 57% on day 2).[17]

**Table A2.** Voting against own signal (estimated by OLS).

| Variable | Coefficient | (Std. Err.) |
| --- | --- | --- |
| Nearby expert with opposite info | −0.034 | (0.079) |
| Neighbors with opposite info | 0.026 | (0.023) |
| Day 1 | −0.021 | (0.031) |
| Row | 0.000 | (0.003) |
| Own signal is blue | 0.044 | (0.030) |
| Green more likely when seeing blue | 0.305 *** | (0.063) |
| Voting blue less likely team decision becomes blue | 0.274 *** | (0.082) |
| Voting blue does not affect team decision | 0.026 | (0.032) |
| Expert info | 0.018 | (0.090) |
| Intercept | 0.111 *** | (0.040) |

Significance levels: * 10%, ** 5%, *** 1%.

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
