# Peer review of "Against All Odds: Tentative Steps toward Efficient Information Sharing in Groups"

_games, doi:10.3390/g11030031_

Round 1

Reviewer 1 Report

Summary:

This article presents the results of a classroom voting experiment with heterogeneous private information about the state of the world. Committees of 35 students would like to implement the true state of the world. Private signals are displayed individually, and with 1/36 chance, one of the subjects in the group is perfectly informed (and she would know so). Two equilibria arise: “Vote your signal” and “let the experts decide” i.e. abstain if you know you are not the perfectly informed. Subjects play this game once, and are then surprised by a second voting decision in which subjects may communicate freely with others prior to receiving any information.

There is an increase in “let the experts decide” strategies between game 1 and 2, meaning that subjects learn about how inefficient it might be to vote given their lack of full information.

Major Comments:

EXPOSITION: the structure of the paper is odd and makes it upleasant to read. Here is what I suggest:

  1. Create a “Model and Theory Section” where you explain the stage game exactly as it will be played in the lab and then provide the propositions/lemmas. Leave all proofs for a supplementary appendix as these are not novel or insightful

  1. “Experimental design section: merge implementation and study design. Make sure you focus only on relevant aspects and send all minor information to a detailed “Session Protocol” online appendix. For example, the body does not need a picture of the classroom. “The presentation also included an animation showing money being distributed to subjects if the majority voted for the true state and this money disappearing from the screen if there is a tie or votes for the true state are in the minority.” This is completely distracting and unnecessary in the body. There are many more instances.

  1. The result section would be better to merge results and robustness as follows: one subsection for the first game, another for the second game and the learning of strategies, and finally the robot programming.

LITERATURE

  1. A key part of your paper is learning and “contagion” of strategies in the lab. Can you cite any relevant papers that have studied this? What about evidence from repeated play in other experiments that you already cite?
  2. Communication: for completeness, you may want to cite literature on communication in other social dilemmas like multilateral bargaining as Baranski and Kagel 2015 JESA or Baranski and Cox (2019)

Minor comments also related to writing:

  1. “In three of the four rounds, the group correctly guessed the majority color in the urn, leading to average earnings of e 50 among subjects selected for payout. “ What do you mean by “in three of the four rounds”? Aren’t there many groups per class room (groups have 35 people)?
  2. The details of the experimental implementation are scattered over all the paper, part in the study design, part in the implementation.
  3. I am not sure the reader needs to know all the exact details of how the experiment was conducted in the body of the paper. Why not have a detailed online appendix for minor aspects?
  4. “Obviously the two days differ in many ways: starting time (whether a lecture is at Monday 7:30 or Tuesday 9:15 is important”. Yes it’s clear that session days differ, but its not clear how these differences imply any differences for your treatments. It’s hard to follow what the argument being made here is. It amounts to saying that “there are differences that may lead to differences in behavior”, which is not really a meaningful statement.
  5. Its impossible to read figure 5 as it stands and comprehend what it conveys. Do you really need it? If so, can it be accompanied by a figure note?
  6. Do you ever mention how many perfectly informed subjects you have in total? I am guessing this is about 5 per session, so it’s a very small sample. This is a major weakness of the study and you should be more transparent about it. Perhaps it’s even more reasonable to argue early on that the unit of study will be the decision of non-experts because of sample size / power issues.
  7. Tables 6,7,8 can be merged into one table and clearly identify each column according to the estimation method. That makes comparisons easier.
  8. In the introduction you write that ““Actual groups typically have some rules or cultures already in place. Hence finding systematic field evidence on the origins of information sharing is difficult. Here, we explore these origins by creating new groups in a situation where people ‘naturally’ differ in their knowledge and understanding: an introductory class with more than 500 students with varying abilities from different study fields. The setting is unique in that students are very well aware of their differences: In class, they have heard both from fellow students, clever and confused questions and answers. To these groups, we give the task to identify whether there are more blue or green balls in an urn.” However, note that student groups also have cultures and roles in place, thus the experiment does not solve this issue. Second, the claim that students know they differ in abilities is true, but these abilities in which they might know they differ have nothing to do with the experimental task per se. Writing and arguments of these type need more care.  

Reviewer 2 Report

Summary: This paper presents the results of a classroom experiment on information aggregation through voting in a setting where, with high probability, one voter is perfectly informed. The paper finds that students do not play the efficient equilibrium of abstaining with low information and voting only with high information. The paper also presents some evidence the students who are physically close to other students who abstain in the first round of the experiment are more likely to abstain in the second round.

Comments: It is unclear to me precisely what research question this study was designed to answer, and why the classroom setting is an ideal setting to answer this question. I have no problem with classroom settings in general, but in this case the lack of control seems to hamper the quality of the results.

First, if the experiment was meant to measure the diffusion of information/coordination on the efficient equilibrium, then a setting with randomized seating would have been much more appropriate. Personally, I find the evidence the authors present that their result is not due to clustering according to type (it's not that I believe the results are not due to information sharing/coordination---it would be quite natural to find this in a classroom setting---but the experiment is not able to identity this effect cleanly).

If this is an experiment to either replicate the findings of earlier papers looking at the swing-voter's curse, then a laboratory setting would facilitate the comparison with earlier papers. And if the authors are interested in a more "real world" setting, then it is unclear that the classroom serves this purpose.

In any case, the authors should identify a novel research question and convincingly show why their experiment is designed to answer this question. We know from other experiments that communication facilitates group's ability to play the efficient NE, and we also know that individuals tend to communicate/vote their signal more than they should relative to a benchmark of rational play. As mentioned earlier, replication studies are always welcome, but it is unclear that this is the purpose of the experiment, and if it was, then the experiment should be designed in precise manner to facilitate the comparison with earlier results.

Round 2

Reviewer 1 Report

Major comment

Please discuss in the text (design and/or conclusion, not in a footnote) the fact that students are not randomly assigned to seats and how this affects the intepretretation/validity of your results. This is a very important point that must be addressed so readers are aware early on and can also cautiously interpret the results.

Author Response

Dear Reviewer 1,

thank you for pointing out the importance of non-random seating for the interpretation of our findings. We now mention non-random seating when discussing its implications for the interpretations of our results in the introduction (middle of page 4). The reader is thus informed early on about what can and cannot be concluded from our results. We found this a good place to mention the issue and hope that you agree.

All the best

Reviewer 2 Report

In response to my earlier review, the authors have updated their introduction and clarified their research question as being:

"Can a group overcome all these challenges and evolve toward the more sophisticated use of information or will it gravitate toward the more primitive use that does not require that everyone fully understands the problem?"

I think the authors would benefit from looking at a much more precise research question---I'm not quite sure how to interpret this question.

Most importantly, however, is that the statement of this research question does little to address my underlying concern that it is unclear what value added a classroom study brings to the literature. If the goal is to bring more real-world elements into the study of information diffusion and information aggregation through voting, then I struggle to see why a classroom setting is more representative of real-world decision making bodies than a laboratory. That is, both are quite far removed from reality, and it is unclear that the classroom setting gets us any closer to real-world decision-making bodies in any dimension that is particularly relevant.

Perhaps the authors should clarify exactly what type of decision-making body they aim to study (referenda? legistlatures? committees of experts?) and what particular dimension of communication they are studying. Again, we know that communication leads to coordination on efficient equilibria, so I would expect either a more novel research question or a replication study.
